# Structure-Based Design of an RNase Chimera for Antimicrobial Therapy

**DOI:** 10.3390/ijms23010095

**Published:** 2021-12-22

**Authors:** Guillem Prats-Ejarque, Helena Lorente, Clara Villalba, Raúl Anguita, Lu Lu, Sergi Vázquez-Monteagudo, Pablo Fernández-Millán, Ester Boix

**Affiliations:** 1Department of Biochemistry and Molecular Biology, Faculty of Biosciences, Universitat Autònoma de Barcelona, 08193 Cerdanyola del Vallès, Spain; helenlorente@hotmail.com (H.L.); clara.villalba@uab.cat (C.V.); Raul.anguita@uab.cat (R.A.); lu.lu@sicau.edu.cn (L.L.); sergi.vazquez96@gmail.com (S.V.-M.); Pablo.fernandez@uab.cat (P.F.-M.); 2College of Animal Science and Technology, Sichuan Agricultural University, Chengdu 625014, China

**Keywords:** RNase, protein engineering, structure-function relationship, antimicrobial proteins

## Abstract

Bacterial resistance to antibiotics urges the development of alternative therapies. Based on the structure-function of antimicrobial members of the RNase A superfamily, we have developed a hybrid enzyme. Within this family, RNase 1 exhibits the highest catalytic activity and the lowest cytotoxicity; in contrast, RNase 3 shows the highest bactericidal action, alas with a reduced catalytic activity. Starting from both parental proteins, we designed a first RNase 3/1-v1 chimera. The construct had a catalytic activity much higher than RNase 3, unfortunately without reaching an equivalent antimicrobial activity. Thus, two new versions were created with improved antimicrobial properties. Both of these versions (RNase 3/1-v2 and -v3) incorporated an antimicrobial loop characteristic of RNase 3, while a flexible RNase 1-specific loop was removed in the latest construct. RNase 3/1-v3 acquired both higher antimicrobial and catalytic activities than previous versions, while retaining the structural determinants for interaction with the RNase inhibitor and displaying non-significant cytotoxicity. Following, we tested the constructs’ ability to eradicate macrophage intracellular infection and observed an enhanced ability in both RNase 3/1-v2 and v3. Interestingly, the inhibition of intracellular infection correlates with the variants’ capacity to induce autophagy. We propose RNase 3/1-v3 chimera as a promising lead for applied therapeutics.

## 1. Introduction

As is largely known, the ribonuclease A superfamily is a vertebrate-specific family of proteins homologous to the bovine pancreatic ribonuclease (RNase A). The proteins of this family share, at the structural level, a high isoelectric point (between 9 and 11), a kidney-shaped structure maintained by three to four disulphide bridges and a common “catalytic triad”, conformed by two histidines and one lysine in the active site, which hydrolyse RNA by an acid-base catalysis [1]. Interestingly, evolutionary studies suggest that the RNase A family started off with a host-defense role [2,3,4]. Antimicrobial and immune-regulatory properties were reported for several family members and their presence in a variety of biological fluids is associated to a protective physiological role [5,6,7,8,9,10].

The antibacterial activity of RNases seems to be unrelated to their ability to degrade RNA [11,12,13,14]. However, recent work is showing that catalytic activity has a role in several aspects of the immune system properties of RNase proteins [15,16,17]. For example, the antiviral and fungicidal activities of RNases seem to be related to the RNase activity [18,19].

Another interesting aspect of RNases, as multifaceted proteins, is their ability to induce autophagy, reported for RNases 3 and 6 [20]. The autophagy induction ability correlates with the protein inhibition of *Mycobacterium aurum* intracellular growth. In accordance, the expression of both RNases in human macrophage cells is up-regulated by the mycobacterial infection, suggesting an in vivo physiological role.

Recently, in our group we have designed a chimeric protein that includes some of the antimicrobial regions of RNase 3 (the RNase with highest antibacterial activity) and the skeleton of RNase 1 (the most catalytically active human RNase). This chimeric protein, named RNase 3/1, presents a considerable antibacterial activity, together with a much higher ribonuclease activity and a lower cytotoxicity in comparison to RNase 3. Not only that, this protein is able to delay the *Acinetobacter baumannii* acquisition of resistance to colistin, when co-administered [21]. In light of these initial promising results, we encouraged ourselves to improve the lead chimera construct and ideated two novel variants with particular ability to target macrophage intracellular infections.

## 2. Results

### 2.1. RNases 3/1-v2 and v3 Rational Design

RNase 3/1 was designed in an effort to combine both the high bactericidal activity of human RNase 3 and the high catalytic activity of human RNase 1, according to previous structural-functional studies [21]. Using the RNase 1 structure as a scaffold, selected antibacterial regions of RNase 3 were added (the cationic regions from Arg77 to Arg81 and Arg103 to Arg107 [11,22], and the N-terminal region of RNase 3 [23,24,25,26]), while retaining the Asp17-Ser26 flexible loop of RNase1, considered essential for RNase1 high catalytic activity [27,28].

Although the RNase 3/1 initial version (thereafter named RNase 3/1-v1) significantly improved the catalytic activity of RNase 3, conserving a significant antimicrobial activity, its bactericidal activity was considerably lower than RNase 3. For this reason, we decided to modify the initial version and introduce cationic and hydrophobic surface residues, proved to be key for the protein antimicrobial properties [11]. With this aim, we added the RNase 3 loop 7 (113N–122Y; thereafter R3-L7) in RNase 3/1-v2 [29] and replaced the flexible loop of RNase 1 (14D–25Y; thereafter R1-L1) by the original shorter loop of the RNase 3 N-terminal (17S–22R; thereafter R3-L1) in RNase 3/1-v3, as shown in Figure 1.

To assess the stability of the designed proteins before their recombinant expression, proteins were modelled, and molecular dynamics was performed with those models. The analysis of the molecular dynamics results shows that the three proteins should be stable, considering both the gyration radius and the backbone RMSD of the proteins (Figure 2). The overall analysis and comparison of the inter-residual distance maps (Appendix A) show conservation of the main structural clusters among the different versions of RNase 3/1. Close inspection of the L1 and L7 loop residues for each variant confirmed that exchange of these surface-exposed regions did not disturb the overall packing of the protein inner core. Likewise, no significant changes are visualized regarding the catalytic triad residues.

The two new versions were successfully expressed in *E. coli* BL21(DE3) cells and purified from inclusion bodies at high yield (between 10 and 40 mg/L, depending on the version). The proper 3D protein folding was checked by circular dichroism (Appendix A), giving the characteristic secondary structure percentage distribution of the RNase A family [33]. Conservation of a proper 3D fold characteristic of the RNase A superfamily was recently confirmed by solving the crystal structure of the three variants. All variant main chain conformation fitted nicely to parental RNases 1 and 3, with an RMSD ranging from 1.2 to 1.7 Å (Fernández-Millán et al., Manuscript in preparation).

### 2.2. Catalytic Activity

Following, we assessed the protein catalytic activity using an activity staining gel. Results are shown in Figure 3. The data indicate that the RNase 3/1-v2 and v3 lowered by half the catalytic activity of the original version towards a polynucleotide substrate, although keeping a remarkable catalytic activity, just 3 times lower than RNase 1. Results indicate that the suppression of the R1–L1 loop does not seem to have a significant effect on the protein catalytic activity.

We also tested the catalytic activity of the different RNase 3/1 versions against dinucleotides (CpA, UpA and UpG) by kinetic spectrophotometry. Results are shown in Table 1. As observed for polynucleotides, there is a slight decrease of the catalytic activity in comparison with the original RNase 3/1 version. Interestingly, the new RNase 3/1 versions appear to have more preference for uridine. Not only that, RNase 3/1-v2 is far more active against adenine than guanidine at the B2 position compared to the other two versions. On the other hand, the suppression of the RNase 1 flexible loop does not seem to have a remarkable effect in the catalytic activity.

### 2.3. Bactericidal Activity and Cytotoxicity

In view of these results, we wanted to check whether the incorporation of the bactericidal loop improved the antimicrobial activity of the RNases 3/1-v2 and 3, as predicted. Therefore, we determined the MBC for the hybrid and parental RNases. Interestingly, the addition of the R3–L7 loop, when the R1–L1 loop is not replaced (being the case of RNase 3/1-v2), results in a decrease of the antimicrobial activity for *A. baumannii* and *P. aeruginosa*, while no improvement of the activity is detected in *E. coli*. In contrast, in RNase 3/1-v3, where the R1–L1 loop of the N-terminal is replaced by R3–L1 together with the R3–L7 loop, a significant improvement of the antimicrobial activity is detected, reaching MBC values of approximately 1 µM, close to RNase 3 activity (Table 2).

Next, the potential cytotoxicity of the three protein versions against host tissues was evaluated in vitro using human tumour hepatic cells (HepG2). The results indicated that RNase 3/1-v2, as observed for v1, shows no toxicity at the maximum concentration tested (150 µM), while RNase 3/1-v3 shows some toxicity at a concentration > 100 µM, although still conserving a good estimated therapeutic index, with MBC_100_ values approximately 100× its calculated IC_50_ toxicity. Results are shown in Table 2. 

Apart from analysing the bactericidal activity against extracellular bacterial strains, we were interested in the activity of the different versions of RNase 3/1 in an infection resistance form, such as an intracellular infection. Therefore, we determined the intracellular MIC values for *Mycobacterium aurum* in RAW macrophages (Figure 4). Results in the macrophage infection model corroborated the enhanced activity of the two new versions in respect to RNase 3/1-v1. Interestingly, in contrast to extracellular MBC results, RNase 3/1-v2 shows similar intracellular antimicrobial activity to RNase 3/1-v3. In contrast, RNase 3/1-v1 shows no detectable activity at the tested concentration range. Results suggest that the addition of the R3–L7 loop, present in both RNase 3/1-v2 and v3, is mandatory for the intracellular antimicrobial activity. However, the two latest versions cannot reach the same levels of activity as RNase 3.

### 2.4. Lipopolysaccharide Binding and Liposome Leakage Activities

Following, we evaluated whether the modifications incorporated in the new versions of RNase 3/1 altered the high LPS-binding affinity previously observed in the first version [21]. LPS-binding affinity was estimated by a displacement assay using the fluorescent BODIPY-cadaverine (BC) probe, as previously described [34]. Results confirmed that the new versions of RNase 3/1 retained a similar LPS-binding affinity, v3 being the one with the highest affinity (Table 3, Appendix A). 

In addition, the liposome leakage ability of the RNases using DOPC/DOPG (3:2) vesicles was evaluated. As shown in Table 4 and Appendix A, RNase 3/1-v3 is the version of RNase 3/1 with the highest liposome leakage activity. After 24 h of incubation, its LC_20_ is identical to RNase 3, with an LC_50_ only slightly lower. Regarding the data at 1 h of exposure, RNase 3/1-v3 shows a remarkably lower leakage activity than RNase 3. These data show that RNase 3/1-v3 is able to disrupt the liposomes at similar levels to RNase 3, although it requires a longer incubation time to reach these levels. On the other hand, RNase 3/1-v1, the version that more closely resembles RNase 1, does not present any significant membrane disruption activity at the registered short time (1 h) exposure, although some leakage activity is observed at 24 h of incubation. Finally, RNase 3/1-v2 has higher liposome leakage activity than RNase 3/1-v1 and lower than RNase 3/1-v3.

### 2.5. Interaction with the RNase Inhibitor (RI)

The presence of the ubiquitous ribonuclease inhibitor (RI) within the cytosol of human somatic cells protects the cells from cellular RNA degradation [35]. The RI binds with a very high affinity to human members of the RNase A superfamily and prevents their cytotoxicity [36]. Its affinity varies depending on the RNase, RNase 1 being the human RNase with the highest affinity with the inhibitor, while RNase 3 presents one of the lowest affinities [35]. Preserving high levels of inhibition by the ribonuclease inhibitor (RI) was one of the main goals of the initial design of RNase 3/1, to ensure host cell protection from the protein’s catalytic activity. Therefore, we evaluated the percentage of RI inhibition of the three RNase 3/1 versions by comparing their catalytic efficiency using CpA as a substrate. As expected, RNase 1 shows total inhibition in the presence of the RI (Appendix A). On the contrary, no significant inhibition could be registered for RNase 3 in the present assay conditions. This might be due to inherent limitations of the assay, where the protein concentration required for RNase 3 to show detectable catalytic activity for the dinucleotide substrate is 5× higher than the other tested proteins, and the corresponding RI volume to be added for full inhibition is slightly above that recommended by the manufacturer. Therefore, the assay allowed us to compare the three variants with RNase 1 but no activity inhibition by the RI could be detected for RNase 3 at the maximum inhibitor/enzyme ratio tested.

In any case, the results highlight the differences between the three variants in relation to RI inhibition of the protein catalytic activities. We observe how RNase 3/1-v2 catalytic activity is mostly not altered by RI, whether a significant reduction of catalytic activity is achieved by RNase 3/1-v1 and RNase 3/1-v3, achieving an inhibition percentage of approximately 70% (Figure 5 and Appendix A). 

### 2.6. RNase 3/1 Autophagy Induction

On the other hand, RNase antimicrobial activity has also been related to diverse immunomodulation activities [20,37]. Interestingly, a screening of human canonical RNases 1–7 in our laboratory highlighted the ability of RNase 3 to induce autophagy in macrophages, which was proven to be essential against intracellular mycobacterial infection [20]. Contrarily, the same experimental results showed that RNase 1 does not activate autophagy.

Therefore, we envisaged here to test whether the hybrid RNase 3/1 chimera conserved the autophagy induction capacity. Towards this aim we assayed the activity of the three RNase 3/1 versions in comparison to RNase 1 and RNase 3 parental proteins. Autophagy induction was evaluated selecting the early and late autophagy markers *BECN1* and *ATG5*, which are essential in the autophagosome formation and maturation, respectively. Results are shown in Figure 6 and highlight that only v2 and v3 retained RNase 3 ability to induce autophagy. Sequence comparison between the first RNase 3/1 chimera and the latter two versions pointed to the R3–L7 loop in RNase 3, which was the only sequence region absent in RNase 3/1-v1 and present in RNase 3 and RNase 3/1 v2 and v3.

Moreover, our previous work disproved that the catalytic activity was involved in the protein autophagy induction activity [20]. Complementarily, we analysed here the main region of RNase 3 associated to its antimicrobial activity: the protein N-terminus [20]. The peptide RN3(1–45) was tested to confirm whether or not this region was involved in the autophagy induction mechanism. In addition, we tested two RNase 3 single mutants at the N-terminus residues previously identified to be key for the protein membrane interaction and cell agglutination activities (residues Ile13 and Trp35) [11,14,38]. The involvement of the N-terminus region was discarded, as the RN3(1-45) peptide could not reproduce the RNase 3 protein induction of autophagy (Figure 7A,B). Besides, single mutations at residues 13 (I13A) and 35 (W35A) did not abolish the protein induction of *BECN1* and *ATG5* expression, as also observed for the catalytically-defective mutant H15A (Figure 7). From these results, we can conclude that the N-terminal 1–45 region is not contributing to the autophagy induction activity. 

In addition, in our previous work we observed that macrophage incubation with RNase 3 was associated with an increase in acidic vesicle formation [20], which could be attributed to the autolysosome compartment during the autophagosome maturation. Here, we inspected whether the RNase 3 ability to aggregate membrane vesicles, a capacity not found in RNase 1, could be responsible for the formation of autolysosomes. Interestingly, the results indicate that the formation of acidic vesicles is not associated to the protein liposome aggregation ability, as the I13A mutant, which is devoid of the aggregation activity [38], is still triggering the increase in acidic vesicles (Appendix A and Appendix A). On the contrary, the protein N-terminus peptide RN3(1-45), which has a positive liposome aggregation activity but cannot induce the autophagy process, is also not able to trigger the formation of acidic vesicles. On the other hand, we discarded the formation of protein amyloid aggregates within the infected macrophages in the assayed conditions (Appendix A). Therefore, we can conclude that no correlation between autophagy induction and liposome or protein aggregation could be inferred from our results. Put together, the analysis of the RT-qPCR and autolysosome formation results indicate that neither the aggregation induction, nor the ribonuclease activity would be responsible for autophagy induction. Instead, the comparison of the three chimera variants indicates that the inclusion of the R3–L7 loop in the RNase 3/1 sequence is essential for autophagy induction.

Seeing these results, we screened the human RNases’ sequences to find out whether the R3–L7 loop included any target region essential for the autophagosome-lysosomal protein routing. By applying a software for the identification of subcellular localization targeting regions, we identified a lysosome tag near the R3–L7 loop (^122^YPVV^125^) in RNase 3 (Appendix A and Appendix A). This region is also present in RNase 1, but it is located in the β6, instead of being exposed in the loop. If we compare the 3D structure of the three RNase 3/1 variants (Fernández-Millán et al., Manuscript in preparation) we can see how the R3–L7 loop of RNase 3 includes the identified autophagosome targeting tag (Figure 8). Specifically, the sequence encompasses not only a lysosome tag, but also an LC3-interacting region (LIR) motif [39,40], with a consensus sequence, [W/F/Y]xx[L/I/V], present at the antimicrobial loop (^122^YPVV^125^). 

## 3. Discussion

RNase 3/1 was designed to achieve a chimeric RNase, which was able to combine, on one hand, the high catalytic activity of RNase 1 and its affinity to the ribonuclease inhibitor, and, on the other hand, the high antimicrobial activity of RNase 3. Following the characterization of the first version of RNase 3/1, which showed its applicability as an antimicrobial protein [21], we decided to improve its activity by designing two new versions of the chimeric protein, RNase 3/1-v2 and RNase 3/1-v3. 

Accordingly, RNase 3/1-v2 was engineered by insertion of an RNase 3 antimicrobial loop (R3–L7), while retaining an RNase 1 loop (R1–L1), essential to providing high catalytic activity. However, structural analysis suggested that the R1–L1 might determine a decrease in the antimicrobial activity of the original RNase 3/1 construct. The L1 loop is placed at the protein N-terminus and links the first two helices (see Figure 1), which were proven essential for RNase 3 antibacterial activity [41]. In RNase 1, the R1–L1 is long and flexible, while in RNase 3, the R3–L1 is a short and rigid loop. As predicted, by replacing the R1–L1 loop with the R3–L1 counterpart in RNase 3/1-v3, we achieved an increase of the protein antimicrobial properties. Previous work in our group attributed the membrane lysis and LPS binding properties of RNase 3 to the first and second α-helices [42]. The present results confirm that the removal of a long and flexible connector between both regions is improving the protein bactericidal ability in RNase 3/1-v3 (Table 2). Interestingly, the molecular dynamics results suggest that the replacement of the R1–L1 loop by the R3–L1 loop induces a structural change that brings the first three N-terminal residues closer to the second helix (see Figure 2 and Appendix A), key for the LPS-binding activity. This different arrangement of the N-terminal, that slightly alters the environment of the two first helices, might explain the improvement in the LPS-binding affinity and antimicrobial activity of the last version of RNase 3/1. Further structural work is ongoing to analyse in depth the structural determinants for RNase 3/1-v3 properties.

More importantly, no significant cytotoxicity is detected at the tested concentrations for any of the three versions of RNase 3/1. However, the outstanding improvement of the antimicrobial activity of v3 chimera in respect to v2 is in detriment of a slight increase of the IC_50_ values calculated for the HepG2 cell line (Table 2). In any case, comparison of MBC_100_ values for the three tested Gram-negative species with the estimated potential toxicity for host cells indicates a ratio well above 100-fold, suggesting a good therapeutic window at the low micromolar range for v3 chimera. 

On the other hand, non-toxicity of RNases has been correlated to their interaction with the RNase inhibitor (RI). The RI, constitutively expressed in most mammalian cells, is considered to protect the cytosol from the RNA degradation by potentially internalized RNases from the RNase A superfamily [43]. Here, we have compared the RI ability to inhibit the catalytic activity of the three chimera variants. Results indicate that the inhibitor slightly reduces its ability to block RNase activity in the protein chimera in comparison to the parental RNase 1 (Figure 5 and Appendix A). Nonetheless, the RI protein retains approximately 70% of its inhibitory activity for RNase 3/1-v1 and approximately 90% for RNase 3/1-v3 chimera. Whereas the insertion of the R3–L7 loop in the case of RNase 3/1-v2 significantly reduces the protein affinity to RI, replacement of the R1–L1 loop with the RNase 3 counterpart would recover appropriate interactions between the N and C-terminus in variant v3. Indeed, structural analysis by circular dichroism confirms that RNase 3/1-v2 presents a higher percentage of random coil than the other two proteins (Appendix A). On the other hand, we should also take into account the clearly differentiated behaviour of both parental proteins, while RNase 1 binds the RI protein at the femtomolar range, RNase 3 has a much lower affinity [36,44]. Interestingly, the data (Figure 5) indicate that RNase 3/1-v3 retains most of the structural elements required for a proper interaction with the RI. Indeed, structural analysis of the three variants and complex binding simulation with the RI (Fernández-Millán et al., in preparation) confirm that the main determinants for interaction are overall retained in the three constructs, the position of most of the key RI residues reported for RNase 1 [45] being conserved in all the variants. Therefore, we can suggest that the decrease of the affinity of RNase 3/1-v2 to the inhibitor is most probably due to an overall structural rearrangement due to the presence of the two long and flexible loops rather than to its specific residue composition. 

Concerning the catalytic activity, the two new versions (v2 and v3) of the protein present a catalytic activity lower than RNase 3/1-v1 against poly(U) and CpA (approximately 40%). Overall, RNase 3/1 chimera achieves a significant enhancement of catalytic activity in respect to RNase 3, between 10 and 50-fold for di- and polynucleotides (Figure 3 and Table 1), although v2 and v3 efficiency is considerably reduced in respect to v1. Interestingly, a careful inspection of chimera activity against dinucleotides (Table 1) reveals significant changes in the protein affinity to main (B1) and secondary (B2) bases. In particular, we observe a shift from cytidine to uridine preference at B1 in the last two variants, which could be attributed to a gradual predominance of RNase 3 traits. Previous kinetic data highlighted the uridine versus cytidine preference of RNase 3 in contrast to RNase 1 [9,46,47]. Further work is currently in progress to analyse the chimera structural peculiarities that explain their differential substrate preferences (Fernández-Millán et al., in preparation).

Another biological property that distinguishes RNase 3 from RNase 1 is its ability to induce autophagy in macrophages, which is correlated to the antimicrobial activity of the protein against intracellular infection [20]. Therefore, we decided here to test the autophagy-inducing ability of our chimeric proteins. Interestingly, while RNase 3/1-v1 has no ability to induce autophagy, the other two versions do present autophagy induction ability (Figure 6). Side by side comparison of chimera sequences identifies the R3–L7 loop, present in both v2 and v3 chimera (Figure 1). Besides, comparative analysis of autophagy markers for the protein N-terminus peptide and RNase 3 mutants (Figure 7) discarded the direct contribution of the 1-45 region in the autophagy induction action. Following, we decided to search for a sequence-specific tag, which might explain the differential behaviour among the three versions of RNase 3/1, and in particular at the R3–L7 region. Sequence screening among the canonical RNases (Appendix A and Appendix A) revealed several subcellular localization tags, mainly associated to lysosomal compartment. Particular interest was drawn by the motif YXXΨ (where Ψ is a hydrophobic residue), found in several RNases, and reported to bind to clathrin associate protein complexes [48,49,50]. However, these data alone cannot explain the observed autophagy induction for RNase 3 and RNase 3/1-v2 and v3. 

To note, this observed tag (^122^YPVV^125^) is not only a stimulator of the clathrin complex, but also an LC3-interacting region (LIR) motif [39,40], where the consensus sequence [W/F/Y]XX[L/I/V] also coincides with the sequence found in the antimicrobial loop (^122^YPVV^125^). Although this tag is present in RNases 1, 3, and the three versions of RNase 3/1, its structural arrangement is distinct for each protein. While in the case of RNase 1 and RNase 3/1-v1 the tag falls into the β6 sheet (Figure 8), in RNase 3 and the other two versions of RNases 3/1, this tag falls within an intrinsically disordered protein region (IDPR) at the R3–L7 loop. As it has been reported recently [40], the functional LIR are short linear motifs which arise from an IDPR. LIR motifs are present in both autophagy receptors and adaptors, which are recognized by Atg8 (LC3) proteins [39,51]. Atg8 proteins are essential for the autophagosome formation, as well as its membrane elongation and the final closure and fusion with lysosomes [52]. In selective autophagy, the Atg8 proteins located on the inner membrane are involved in the recruitment of the autophagosome cargo by binding to autophagy receptors [39,52]. Interestingly, the sequence found in RNase 3 is a Y-type motif (i.e., a motif with a tyrosine at the beginning of the consensus sequence). The Y-type motifs are reported to be the less common of the LIR motifs [39], and the ones with the lowest affinity with LC3 proteins [53]. Thus, the RNase 3 recruitment by Atg8 proteins would be achieved at a lower ratio than autophagy receptors. What could be the role then of the RNase recruitment into the autophagosome cargo? The presence of RNases in the autophagosome could be involved in the killing of the pathogen inside the autophagosome, as well as the removal of the pathogenic RNA. Indeed, recent work reported that RNase 2 (which presents the same motif in the same region as RNase 3, see Figure 8), colocalized with the early and late endosomal markers, Rab5 and Rab7, together with the endolysosomal marker LAMP1 [54]. This colocalization at the endolysosomal level would enable the TLR8-induced immune response, triggered by the local degradation of pathogen RNA [54,55].

In light of these data, taking into account that the activation of the autophagy process does not seem to be associated with the RNase catalytic activity (Figure 7) [20], it would be tempting to conclude that the LIR motif is the one responsible for the autophagosome recruitment. We can speculate that the RNase recruited into the autophagosome could help to remove the bacteria inside the vesicle. 

Put together, all these data allow us to conclude that the presence of the R3–L7 loop in RNase3 and -v2 and -v3 chimera correlates with the protein antibacterial activity, not only because of its higher cationicity, as previously identified [11], but also because of its ability to induce the autophagy process. Accordingly, when we assessed the intracellular MIC in macrophages infected with *Mycobacterium aurum*, only RNase 3 and RNases 3/1-v2 and v3 variants, which present the LIR tag, were active (Figure 4). The fact that RNase 3/1-v1, which has higher bactericidal activity against extracellular bacteria than RNase 3/1-v2, does not show any detectable activity at the intracellular level, suggests to us the importance of this LIR tag in the cellular mechanisms for removal of bacterial infection. Further studies will be undertaken to elucidate the participation of this tag into the bactericidal mechanism of RNases. 

Our previous work reported the successful use of an RNase 3/1 chimera against the emergence of bacterial resistance in an *Acinetobacter baumannii* culture exposed to colistin [21], where catalytic activity is essential (Li et al., in preparation). Here we have developed two novel variants combining catalytic activity with enhanced antimicrobial properties (Figure 9) that reinforce the potential of RNase-base compounds as alternative antimicrobial drugs.

## 4. Materials and Methods

### 4.1. Materials

The *Acinetobacter baumannii* strain (CECT 452; ATCC 15308) and *Pseudomonas aeruginosa* strain (CECT 4122; ATCC 15692) are from the Spanish Type Culture Collection (CECT). The *Escherichia coli* BL21(DE3) strain and the pET11c plasmid are from Novagen. The HepG2 cells are from the American Type Cell Culture Collection (ATCC HB-8065). The MEM medium is from Lona and FBS is from Gibco. CpA, UpA and UpG are from IBA Life Sciences. The Ribonuclease inhibitor (RNasin^®^) is from Promega. Poly(C), LPS (0111:B4 serotype) and RNase A (Type XII) are from Sigma-Aldrich. 3-[4,5-dimethylthiazol-2-yl]-2,5-diphenyl tetrazolium bromide (MTT), isopropyl β-D-1-thiogalactopyranoside (IPTG), and colistin are from Apollo Scientific (Bredbury, Chesire, UK). 1-aminonaphthalene-3,6,8-trisulfonate (ANTS), α,α′-dipyridinium p-xylene dibromide (DPX), and the fluorescent probe BODIPY TR cadaverine are from Molecular Probes (Eugene, OR, USA). Toluidine blue is from Merck (Kenilworth, NJ, USA). The RNase3/1 gene was purchased from NZYTech (Lisboa, Portugal). 1,2-dioleoyl-*sn*-glycero-3-phosphocholine (DOPC) and 1,2-dioleoyl-*sn*-glycero-3-phosphoglycerol (DOPG) were purchased from Avanti Polar Lipids (Birmingham, West Midlands, UK). The RN3(1-45) peptide [24] was kindly synthetized by the Peptide Synthesis Core Facility (Universitat Pompeu Fabra, Barcelona, Spain, Prof. David Andreu Laboratory). The human RNase 1 gene was a gift from Dr. Maria Vilanova, Universitat de Girona, Girona, Spain. The human RNase 3 sequence was taken from a previously synthesized gene [44].

### 4.2. Protein Expression and Purification

The RNase 1, 3, and 3/1 genes were subcloned into the plasmid pET11c for prokaryote high yield expression in an *E. coli* BL21(DE3) strain. The recombinant protein was expressed and purified as previously described [46], with some modifications [56]. Briefly, bacteria were grown in Terrific broth (TB), containing 400 μg/mL ampicillin. Recombinant protein was expressed after cell induction with 1 mM IPTG added when the culture showed an OD_600_ of 0.6. The cell pellet was collected after 4 h of culture at 37 °C. Cells were resuspended in 10 mM Tris/HCl and 2 mM EDTA, pH 8 and 40 μg/mL of lysozyme, and sonicated after 30 min. The pellet was suspended in 25 mL of the same buffer with 1% triton X-100 and 1 M urea and was left stirring at room temperature for 30 min, before being centrifuged for 30 min at 22,000× *g*. This procedure was repeated until the supernatant was completely transparent. In order to remove the triton X-100, 200 mL of 10 mM Tris-HCl pH 8.5, 2 mM EDTA was added to the pellet and centrifuged again for 30 min at 22,000× *g*. The resulting pellet was suspended in 25 mL of Tris-acetate 100 mM, pH 8.5, 2 mM EDTA, 6 M guanidine hydrochloride, and 80 mM of DTT. The protein was then refolded for 72 h at 4 °C by a rapid 100-fold dilution into 100 mM Tris/HCl, pH 8.5, 0.5 M of guanidinium chloride, and 0.5 M L-arginine, and oxidized glutathione (GSSG) was added to obtain a DTT/GSSG ratio of 4. The folded protein was then concentrated, buffer-exchanged against 150 mM sodium acetate, pH 5, and purified by cation-exchange chromatography on a Resource S (GE Healthcare) column equilibrated with the same buffer. The protein was eluted with a linear NaCl gradient from 0 to 2 M in 150 mM sodium acetate, pH 5.

### 4.3. Mutagenesis and Construction of RNase 3/1 Chimera

The two new versions of RNase 3/1 were engineered starting from the original gene of RNase 3/1 [21]. To generate RNase 3/1-v2, the sequence corresponding to RNase3 loop 7 (R3-L7) was added using the primers 5′AATCGTGATCCCCGTGATAGTCCCCGTTACCCGTATGTGCCGGTGCATTTTG 3′ (forward) and 5′ CTCGCATGCCACGATGATATGG 3′ (reverse). The lineal product was amplified with *Supreme NZYProof polymerase* (NZYTech), digested with *DpnI* and cleaned with *GenElute PCR Clean-Up kit™* (Sigma Aldrich, Burlington, VT, USA), and then phosphorylated with T4 PNK and circularized by Anza T4 DNA ligase (Invitrogen, Waltham, MA, USA). The procedure for the construction of RNase 3/1-v3 was the same but using RNase 3/1-v2 as a template and the primers to replace the R1–L1 loop to the R3–L1 loop: 5′ AGCCTGAACCCGCCGCGCTGCACCATTGCAATGCG 3′ (forward) and 5′ GATATGTTGAATCGCAAACCAC 3′ (reverse). 

### 4.4. Molecular Dynamic Simulations

To analyse the stability of the new versions of RNase 3/1, molecular dynamics were performed. All the molecular dynamics were performed with GROMACS 2018.1 [57], using *AMBER99SB-ILDN* [58] as a force field. Proteins were centred in a dodecahedral box with a distance of 1 nm between the box and the solvent. The unit cell was filled with *TIP3P* water [59] at a neutral pH and 100 mM of NaCl. 

For the neighbour search a Verlet cut-off scheme was used [60] with a cut-off of 0.9 nm for both Van der Waals and short-range Lennard-Jones interactions. For the long-range interactions, smooth particle mesh of Ewald (PME) [61,62] was used with a fourth-order interpolation scheme and 0.1125 nm grid spacing for FFT. The bonds were constrained with P-LINCS algorithm [63] with an integration time step of 2 fs.

The energy of the systems was minimized using the steepest descendant algorithm [64] and equilibrated in two steps. First, an initial constant volume equilibration (NVT) of 1 ns was performed with a temperature of 298 K using a Berendsen modified thermostat [65]. Then, 1 ns of constant pressure equilibration (NPT) was run at 1 bar with a Parrinello-Rahman barostat [66,67] at 298 K and the same thermostat. Finally, 50 ns production runs were performed under an NPT ensemble without applying restraints. 

### 4.5. Circular Dichroism (CD)

Far-UV CD spectra were obtained from a Jasco-715 (Jasco), as previously described [24]. The spectra were registered from 195 to 240 nm at room temperature. Data from 4 consecutive scans were averaged. Before reading, the sample was centrifuged at 10,000× *g* for 5 min. Spectra of RNase 3/1 were obtained at 6 µM in 5 mM sodium phosphate, pH 7.5, with a 0.2 cm path-length quartz cuvette. The percentage of secondary structure was estimated with Spectra Manager II, as described [68]. 

### 4.6. Activity Staining Gel

Zymograms were performed following the method previously described [69]. Fifteen percent polyacrylamide-SDS gels were cast with 0.3 mg/mL of poly(C) (Sigma Aldrich, Burlington, VT, USA). Then, 20 ng of RNase 1, 3, and 3/1 were loaded, and the gel was run at a constant current of 100 V for 1.5 h. Following, the SDS was removed from the gel with 10 mM Tris/HCl, pH 8, and 10% (*v/v*) isopropanol. The gel was then incubated during 1 h in the activity buffer (100 mM Tris/HCl, pH 8) to allow enzymatic digestion of the embedded substrate and then stained with 0.2% (*w/v*) toluidine blue in 10 mM Tris/HCl, pH 8, for 10 min. Positive bands appeared white against the blue background. The loading buffer had no 2-mercaptoethanol to facilitate recovery of active enzymes. 

### 4.7. Minimum Bactericidal Concentration (MBC) Determination

MBC was defined as the lowest protein/peptide concentration that completely eradicated bacterial cells. RNase 3/1 was serially diluted from 20 to 0.02 μM in HBS (HEPES 20 mm pH 7.4, NaCl 100 mM). Then, an exponential phase subculture of *E. coli*, *A. baumannii,* or *P. aeruginosa* was added, previously adjusted to give a final concentration of approximately 5·10^5^ colony-forming units (CFU)/mL in each well and the plate was incubated for 4 h at 37 °C and 100 rpm. Finally, samples were plated onto LB (Condalab, Madrid, Spain) Petri dishes and incubated at 37 °C overnight. All the assays were performed in triplicate.

### 4.8. Cytotoxicity Assay

Cytotoxicity was measured for HepG2 human cell lines using the MTT assay, as described previously [42]. Cells were grown in 5% CO_2_ at 37 °C with minimal essential medium supplemented with 10% fetal bovine serum (FBS). Cells were plated at 5 × 10^4^ cells/well in a 96-well plate and incubated overnight. Next, the medium was removed, and serial dilutions of proteins were added at concentrations ranging from 200 to 0.2 μM in 100 μL of medium without serum. After 4 h of incubation, the medium was replaced with fresh medium containing 0.5 mg/mL MTT solution and the mixture was incubated for 2 h in 5% CO_2_ at 37 °C. The medium was then removed, and formazan was dissolved by adding acidic isopropanol. The optical density (OD) was recorded by using a Victor3^TM^ plate reader (PerkinElmer, Waltham, MA, USA) set at 550 nm and 630 nm as references. Reference absorbance at 630 nm was used to correct for nonspecific background values. Each experiment was repeated at least 3 times.

### 4.9. LPS Binding Assay

The LPS-binding affinity was assessed using the fluorescent probe BODIPY TR cadaverine (BC) as described [34]. Proteins were serially diluted in a 96-well fluorescence plate from 20 to 0.02 μM in HEPES 10 mM pH 7.4. Then, LPS (10 µg/mL) and BC (10 µM) were added in the same buffer. Fluorescence measurements were performed on a Victor3 plate. The BC excitation wavelength was 580 nm, and the emission wavelength was 620 nm. The occupancy factor was calculated as described previously [34].

### 4.10. Liposome Preparation

Large unilamellar vesicles (LUVs) containing DOPC/DOPG (3:2 molar ratio, 1 mM stock concentration) of a defined size were obtained from a vacuum drying lipid chloroform solution. After the chloroform evaporation, liposomes were suspended with 10 mM Tris/HCl, 20 mM NaCl, pH 7.4. The distribution and the mean hydrodynamic range of the liposomes in suspension was determined by dynamic light scattering (DLS) with a Zetasizer Nano ZS Malvern, and the data were analysed by its built-in software (Zetasizer 7.02, Malvern Panalytical, Malvern, Worcershire, UK).

### 4.11. Liposome Leakage

The ANTS/DPX liposome leakage fluorescence assay was performed as previously described [70], with slight modifications. Briefly, a unique population of LUVs DOPC/DOPG (3:2) liposomes was prepared to encapsulate a solution containing 12.5 mM ANTS, 45 mM DPX, 20 mM NaCl, and 10 mM Tris/HCl, pH 7.5 with 3 freeze-thaw cycles. The liposomes were later extruded through a 100 nm membrane and the free fluorophores were eliminated via gel filtration chromatography. The ANTS/DPX liposome stock suspension was diluted to 100 µM and incubated at room temperature with RNases 1, 3, and the three versions of RNase 3/1, serially diluted from 50 to 0.024 µM in an opaque microtiter fluorescence plate. SDS at 5% was taken as a positive control to determine the 100% of leakage. Fluorescence measurements were performed after 1 h and 24 h of incubation on a Victor3 plate reader with an excitation wavelength of 405 nm and an emission wavelength of 535 nm. LC_20_ and LC_50_ values were calculated by fitting the data to a dose-response curve with GraphPad Prism 9. The assay was performed in duplicate.

### 4.12. Spectrophotometric Kinetic Analysis

The catalytic activity of the proteins against CpA, UpA and UpG was tested by a spectrophotometry assay, as previously described [71]. Assays were carried out in 50 mM sodium acetate and 1 mM EDTA, pH 5.5, at 25 °C, using 1-cm path length quartz cells. Substrate concentration was determined spectrophotometrically using the following molar absorption coefficients: ε_265_ = 21,000 M^−1^·cm^−1^ for CpA, ε_261_ =23,500 M^−1^·cm^−1^ for UpA, ε_261_=20,600 M^−1^·cm^−1^ for UpG. The activity was measured at 286 nm wavelength for both CpA and UpA and 280 for UpG. The activity was measured by following the initial reaction velocities using the different molar absorbance coefficients, in relation to the cleaved phosphodiester bonds. The relative activity was calculated by comparison of initial velocities (V_0_), using a substrate concentration of 0.1 mM for the 3 dinucleotides.

### 4.13. Ribonuclease Inhibitor Activity Assay

The ribonuclease activity of RNase 1, 3, and the three variants of RNase 3/1 in the presence or absence of ribonuclease inhibitor (rRNasin^®^, Promega, Madison, WI, USA) was assayed as previously described [72], with slight modifications. The buffer for the reaction was 50 mM MES-NaOH pH 6, 125 mM NaCl, 1 mM EDTA, 1.2 mM DTT, 0.1% PEG 4000, and 0.2 mg/mL BSA. Proteins were incubated in the reaction buffer, in the presence or absence of the RNase inhibitor (RI), for 30 min at room temperature to facilitate binding to the inhibitor prior to the reaction. The enzymatic reaction was carried out in 70 μL UV-compatible disposable plastic cuvettes with 1 cm of optical pathlength. The substrate used was CpA at 100 μM. The protein concentrations used were 0.2 µM for RNase 1 and the three versions of RNase 3/1, and 1 μM for RNase 3; the inhibitor was added at a ratio of 5 inhibitor units per 100 nM of protein. The catalytic activity against the dinucleotide CpA was measured as a decrease in absorbance at 286 nm, which corresponds to the first step of the phosphodiester bond cleavage.

### 4.14. Macrophage Cell Culture, Infection and Intracellular MIC Determination

Macrophage cell culture and infection was performed as previously described (Lu et al., 2019) with some modifications. Mouse RAW 264.7 cells (NCTC, #91062702) were maintained or passaged in 25 cm^2^ tissue culture flasks (BD Biosciences, 353108) using a DMEM (Lonza, BE04-687F/U1) and RPMI-1640 (Lonza, BE12-702F) medium with 10% heat-inactivated fetal bovine serum (FBS, 26140079, Gibco, Waltham, MA, USA), respectively, at 37 °C and humidified under 5% CO_2_ conditions. RAW 264.7 cells were seeded at 5 × 10^5^ cells/well and allowed to attach for 2 h before infection and treatment. The number of viable cells was counted using a Trypan blue (Invitrogen, 15250-061) exclusion assay.

To determine the intracellular Minimum Inhibitory Concentration (MIC), RAW264.7 macrophages were infected with *Mycobacterium aurum*. First, cells were plated at 5·10^5^ cells/well in 24-well plates and after 2 h of adhesion, the medium was removed and DMEM + FBS with *M. aurum* at a multiplicity of infection (MOI) of 10:1 was added. Bacterial concentration was previously adjusted spectrophotometrically at an OD_600_ = 1, when the culture is at the log-phase. Then, ultrasound sonication was used to ensure proper bacterial resuspension in the media. After that, macrophages were incubated for 3 h with the infection medium to allow bacterial internalization [20]. Then, the macrophages were washed 3 times with PBS and treated with RNases at the corresponding concentration for 24 h in a medium containing 50 µg/mL gentamycin (to remove extracellular bacteria). Once the treatment was completed, the macrophages were washed 2 times with PBS and lysed using 500 μL of distilled water. Cell lysates were concentrated to 50 μL by centrifugation, plated into MB7H10/OADC agar plates and incubated at 37 °C for 4–5 days for subsequent colony counting.

### 4.15. Real-Time qPCR Assays

Total RNA of the mouse macrophages was extracted using *mirVana™ miRNA Isolation Kit* as described by the manufacturer (AM1560, Ambion, Life Technologies, Waltham, MA, USA) at each time point (4, 24, 48, 72 h). Total RNA was quantified by NanoDrop™ spectrophotometer (Thermo Fisher Scientific, Wilmington, DE USA), and cDNA was synthesized using iScript^TM^ cDNA Synthesis Kit (Bio-Rad, 170-8891). RT-qPCR was performed as previously described using the corresponding primers for mouse *BECN1* and *ATG5* [20]. Mouse *BECN1* and *ATG5* genes relative to mouse actin (as a housekeeping gene) were measured in triplicate from cDNA samples by real-time quantitative PCR using CFX96 Real-Time PCR detection system (Bio-Rad, Hercules, CA, USA). The results were analysed by using the relative standard method [73].

### 4.16. Acridine Orange Staining

To quantify autolysosome formation, mouse macrophage cells treated with proteins or peptide were stained with acridine orange as previously reported [74]. Briefly, cells were seeded in 96-well plates (10^4^ cells/well) and treated with RNases and the RN3(1–45) peptide. At the end of the treatments, cells were washed with prewarmed PBS and stained with 5 µg/mL acridine orange (AO) for 10 min in PBS. After washing with PBS 3 times, autolysosome formation was performed measuring the red/green fluorescence intensity ratio of AO staining (AO green fluorescence λ_exc_ 485 nm and λ_em_ 535 nm; AO red fluorescence λ_exc_ 430 nm and λ_em_ 590 nm). Values were normalized on cell proliferation by MTT assay.

### 4.17. Intracellular Aggregates Formation Measurement by Thioflavin-S Staining 

Mouse macrophage cells treated with protein or peptide were stained with Thioflavin-S [75]. Briefly, cells were seeded in 96-well plates (10^5^ cells/well) and treated with RNases or the RN3(1-45) peptide. At the end of the treatments, cells were washed with prewarmed PBS and stained with 25 µM of Thioflavin-S for 15 min in PBS. After washing with PBS 3 times, fluorescence intensity was recorded under condition of λ_exc_ 375 nm and λ_em_ 455 nm.

## 5. Conclusions

In this work, we designed and characterized RNase 3/1 chimera. Based on a first RNase 3/1 chimera, two new versions (RNases 3/1-v2 and v3) incorporate a C-terminal loop key for the protein antimicrobial properties. In particular, the -v3 variant lacks an N-terminal loop that provides local flexibility in RNase 1 but may induce some reshaping of the overall conformation in the RNase 3/1 chimera due to potential steric clashes between the N and C-terminus regions. The RNase 3/1-v3 shows enhanced antimicrobial and catalytic activities in relation to v2, while retaining equivalent RI interaction to RNase 3/1-v1 and non-significant toxicity for the tested human cell line. Interestingly, we have found an LIR motif in RNase 3, which is present in the RNase 3/1-v2 and v3 versions and is associated with the autophagosome formation. The presence of an LIR tag correlates with both the protein autophagy induction and eradication of macrophage intracellular infection.

## Figures and Tables

**Figure 1 ijms-23-00095-f001:**
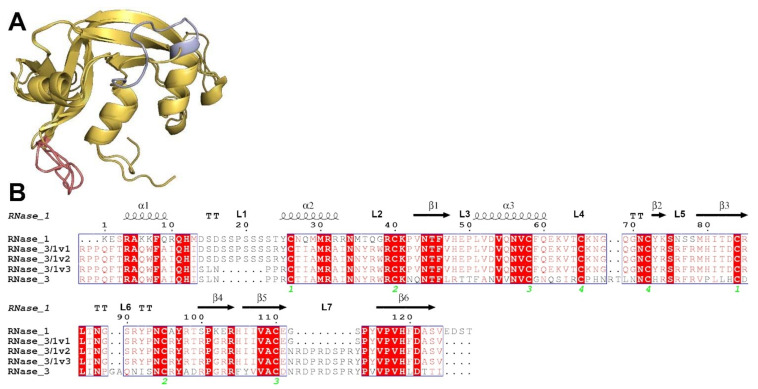
(**A**) Overlapping of the models of RNase 3/1-v1-3 obtained by Modeller 9.12 [30]. In red, the antimicrobial loop of RNase 3, added in RNase 3/1-v2. In blue, the flexible loop of RNase 1, removed in RNase 3/1-v3. In beige, the RNase 1 skeleton. (**B**) Alignment between RNases 1, 3 and the three versions of RNase 3/1. The fully conserved amino acids are highlighted in red. The residues that are not conserved but have similar properties are marked with red letters. The secondary structure is indicated above the alignment. The green numbers indicate the disulphide bridges. The alignment was conducted using Clustal Omega [31], and the image was obtained with ESPript7 (http://espript.ibcp.fr/ESPript/) [32].

**Figure 2 ijms-23-00095-f002:**
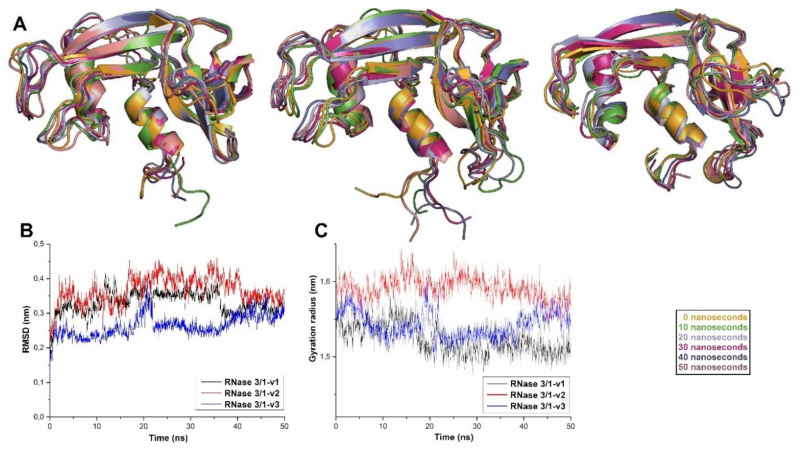
(**A**) Overlapping of the three versions of RNase 3/1 after 50 nanoseconds of molecular dynamics. (**B**) RMSD variation for the three hybrid RNases. (**C**) Gyration radius of the structure of hybrid RNases against time.

**Figure 3 ijms-23-00095-f003:**
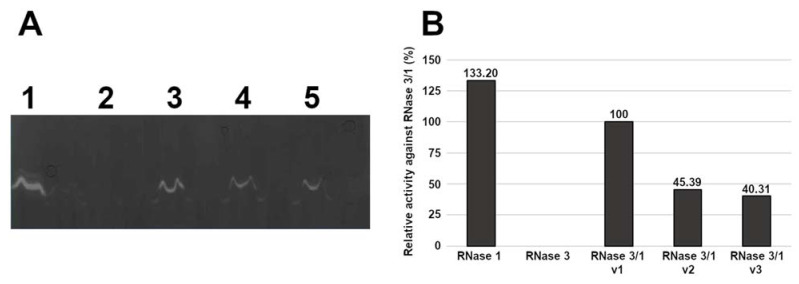
(**A**) Activity staining gel showing the degradation of poly(U) with 15 ng of each protein. RNase 1 (lane 1), RNase 3 (lane 2), RNase 3/1-v1 (lane 3), RNase 3/1-v2 (lane 4) and RNase 3/1-v3 (lane 5). (**B**) Densitometric analysis of the bands of the activity staining gel. Activity is represented as relative activity compared to RNase 3/1-v1. Quantification was conducted from a scan image using Quantity One software (Bio-Rad^®^,Hercules, CA, USA).

**Figure 4 ijms-23-00095-f004:**
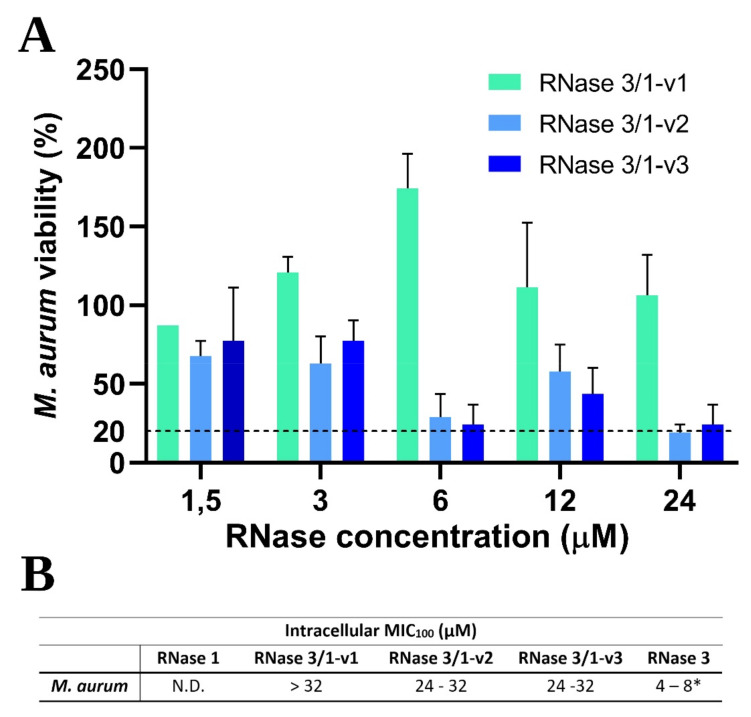
Intracellular *Mycobacterium aurum* viability in infected macrophages after 24 h of treatment with the proteins at the indicated concentrations. (**A**) *M. aurum* viability at each concentration of protein for the three versions of RNase 3/1. The horizontal bar shows the 20% of viability level. (**B**) The embedded table shows the intracellular MIC_100_ values of the RNase 3/1 versions in comparison with its parental proteins. N.D., not detected at the assayed conditions; MIC_100_ values of the three versions. * RNase 3 MIC value taken from [20].

**Figure 5 ijms-23-00095-f005:**
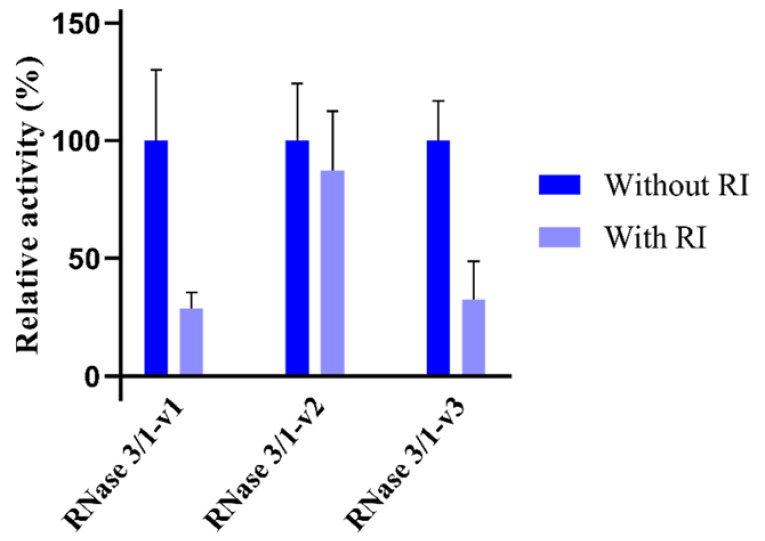
Ribonuclease inhibitor (RI) percentage of inhibition was determined by comparing the catalytic activity against CpA in the presence or absence of the inhibitor. Values represent the percentage of inhibition. The calculated activity in µmols of product per µmol of enzyme is shown in Appendix A. Results are shown from at least three replicates (mean ± SEM).

**Figure 6 ijms-23-00095-f006:**
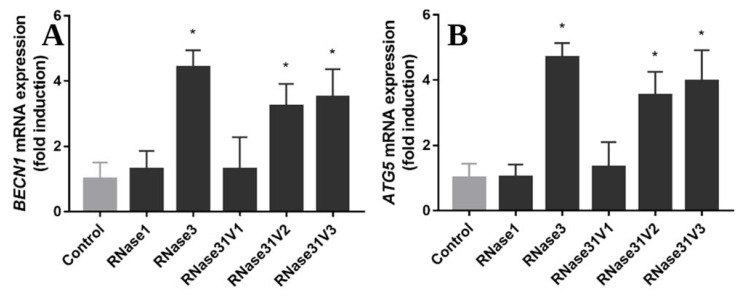
RAW 264.7 macrophages were infected with *M. aurum* and treated by 10 µM of proteins (RNase 1, 3 and 3/1 v1-3) for 24 h. Real-time qPCR measured the relative expression of *BECN1* (**A**) and *ATG5* (**B**) genes, normalized by housekeeping gene *β-actin*. Results are shown from three independent experiments (mean ± SD). * indicates significant difference compared with control group (*p* < 0.05).

**Figure 7 ijms-23-00095-f007:**
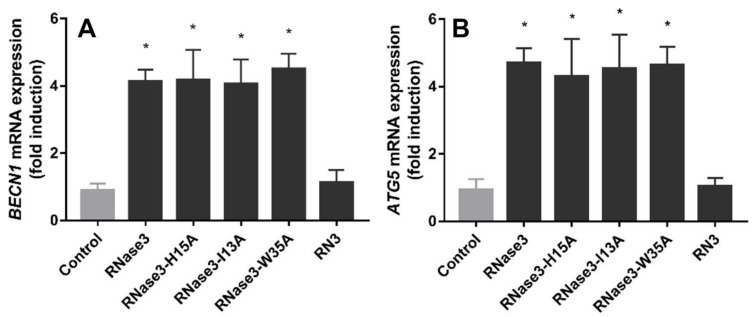
RAW 264.7 macrophages were infected with *M. aurum* and treated with 10 µM of protein or peptide (RNase 3, its mutants I13A, H15A and W35A, and the peptide RN3(1-45)) for 24 h. Real-time qPCR measured the relative expression of *BECN1* (**A**) and *ATG5* (**B**) genes, normalized by housekeeping gene *β-actin*. Results are shown from three independent experiments (mean ± SD). * indicates significant difference compared with control group (*p* < 0.05). RNase 3-H15A data are taken from [20].

**Figure 8 ijms-23-00095-f008:**
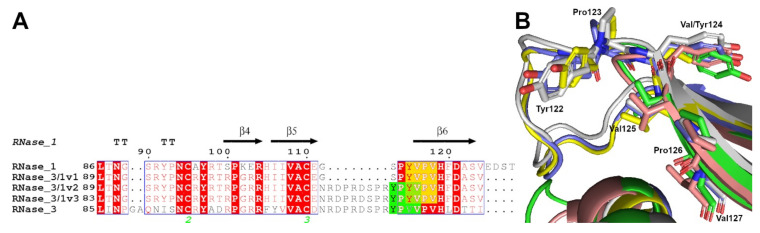
(**A**) Sequence alignment of the C-terminal of RNases 1, 3 and the three versions of RNase 3/1. In yellow the sequence tag which, being in the β6 sheet, does not act as a LIR tag. In green, the LIR tag. (**B**) Zoom of the putative LIR-AIM tag found. In salmon, RNase 1 (2k11.pdb); in yellow, RNase 3 (1qmt.pdb); in green, RNase 3/1-v1 (6ymt.pdb); in grey, RNase 3/-1-v2 (6ybc.pdb); in blue, RNase 3/1-v3 (6ssn.pdb). The residues of the putative tag are shown. Sequence numbering is according to the RNase 3. Picture was obtained by *PyMOL 2.3.4*, (Schrödinger LLC, New York, NY, USA).

**Figure 9 ijms-23-00095-f009:**
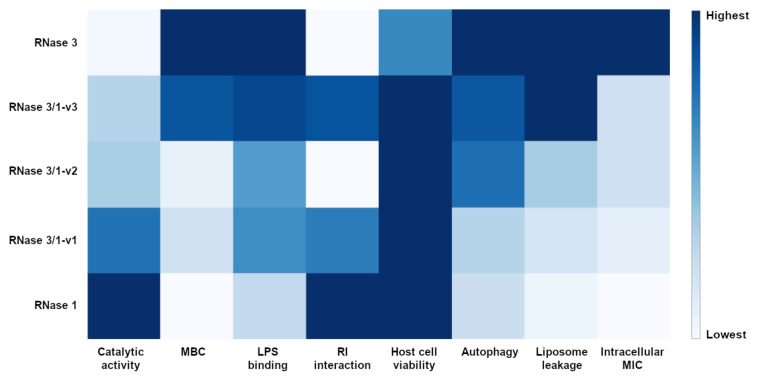
Heatmap depiction for activity comparison of RNase 1, 3 and the three versions of RNase 3/1. All the results were normalized as relative to the highest detected activity and the heatmap was depicted by Plotly and GIMP.

**Table 1 ijms-23-00095-t001:** Comparison of the relative catalytic activity against dinucleotides. Values of relative activity are calculated from the average of three replicates, considering RNase 3/1-v1 as 100% of activity.

Relative Catalytic Activity against Dinucleotides					
	RNase 1	RNase 3	RNase 3/1-v1	RNase 3/1-v2	RNase 3/1-v3
**CpA**	641	22	100	37	35
**UpA**	940	1	100	73	130
**UpG**	124	N.D.	100	23	85

**Table 2 ijms-23-00095-t002:** Bactericidal and cytotoxic activities of RNases 1, 3 and the three versions of RNase 3/1. Bactericidal activity (MBC_100_) against three Gram-negative bacterial species was determined as the lowest concentration of protein where no cells were detected by CFU counting in HBS. Cell toxicity (IC_50_) was determined as the concentration of protein where 50% of cells were still alive. Cytotoxicity was determined by the MTT assay. N.D. means that, at the highest tested concentration (150 µM), no reduction of cell viability was detected. ^1^Cell toxicity values of RNase 3/1-v1 are taken from [21]. Each assay was performed in triplicate.

MBC100 (µM)					
	RNase 1	RNase 3/1-v1	RNase 3/1-v2	RNase 3/1-v3	RNase 3
** *E. coli* **	>20	6.25 ± 2.17	6.25 ± 2.17	0.78 ± 0.27	1.88 ± 0.88
** *A. baumannii* **	>20	6.25 ± 2.17	16.67 ± 2.89	1.56 ± 1.89	0.6 ± 0.07
** *P. aeruginosa* **	18.33 ± 2.89	3.13 ± 1.08	18.33 ± 2.89	1.25 ± 0.54	0.6 ± 0.07
**Cell Toxicity (IC50) (µM)**					
	RNase 1	RNase 3/1-v11	RNase 3/1-v2	RNase 3/1-v3	RNase 3
**HepG2 cells**	N.D.	N.D.	N.D.	N.D.	134.66 ± 0.95

**Table 3 ijms-23-00095-t003:** LPS-binding affinity and liposome leakage activity of RNases 1, 3 and 3/1. The LPS concentration that displaced 50% of the bound fluorescent probe (LBC_50_) has been determined from a dose-response curve adjusted by OriginPro *8* statistical software (Origin Lab, Northampton, MA, USA). Experimental data plots are shown in Appendix A.

LPS Binding (LBC_50_) (µM)				
RNase 1	RNase 3/1-v1	RNase 3/1-v2	RNase 3/1-v3	RNase 3
1.50 ± 0.28	0.59 ± 0.02	0.66 ± 0.08	0.42 ± 0.07	0.38 ± 0.03

**Table 4 ijms-23-00095-t004:** Liposome leakage activity of RNases 1, 3 and the three versions of RNase 3/1. DOPC/DOPG (3:2) liposomes were prepared as described in the methodology. The values in parentheses represent the 95% confidence interval. Liposome leakage was measured as the protein concentration that induces 20% or 50% of liposome leakage (LC_20_ and LC_50_, respectively), using a dose-response curve adjusted by GraphPad Prism 9 (Graph Pad, San Diego, CA, USA). Experimental data plots are shown in Appendix A. N. D.: not detected at the concentrations tested.

	Liposome Leakage			
	1 h Exposure		24 h Exposure	
	LC_20_ (µM)	LC_50_ (µM)	LC_20_ (µM)	LC_50_ (µM)
**RNase 1**	42.73	N. D.	16.8	N. D.
	(27.02–57.29)		(4.28–37.31)	
**RNase 3/1-v1**	N. D.	N. D.	5.766	N. D.
			(2.19–15.33)	
**RNase 3/1-v2**	24.33	34.19	2.916	12.35
	(21.39–27.42)	(31.39–37.29)	(2.13–3.9)	(10.03–15.36)
**RNase 3/1-v3**	26.74	60.19	1.028	4.541
	(22.89–30.87)	(52.39–73.5)	(0.43–2.07)	(2.7–8.07)
**RNase 3**	5.53	12.85	1.027	1.682
	(4.17–7.17)	(10.92–15.09)	(0.49–1.66)	(1.14–2.35)

## Data Availability

Not applicable.

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
