# Peer review of "Structure-Based Design of an RNase Chimera for Antimicrobial Therapy"

_ijms, 2021, doi:10.3390/ijms23010095_

Round 1
Reviewer 1 Report
I believe it is a well designed research paper explaining in a clear way their new proposition in antimicrobial therapeutics
Author Response
We sincerely thank the reviewer for the positive evaluation of our manuscript.
Reviewer 2 Report
The protocol for dynamics simulation is correct.
However the final results is lacking the easy interpretation. This is why the following presentation is suggested:
the distance map visualising the inter-rrsidual changes for discussed forms of ribonuclease. The distance map shall visualise the pairs of positions with incresed and decreased distances. The activity-related positions shall be also distinguished on the map as well as conserved positions.
This map shall be the suplement to 3D presentations which are difficult to follow the structural changes.
Author Response
We thank the reviewer for the manuscript revision and suggestions. Reviewer is right that the 3D representation does not allow to fully appreciate the main differences between the three variants. As suggested, we have prepared an inter-residual distance map for each of the RNase 3/1 chimera, where it is possible to get an overall comparison and visualize the specific interactions between residues. Also, for a better clarity we have added the secondary structure labels and highlighted the key loops (L1 and L7) related to the constructs’ biological properties, together with identification of the active site residues (see Figure S1). Accordingly, the results and discussion sections have been updated to introduce the new information analysis.